

# 1 Modelling forest plantations for carbon uptake
# 2 with the LPJmL dynamic global vegetation
# 3 model

*Maarten C. Braakhekke[1,2,*], Jonathan C. Doelman[2], Peter Baas[3], Christoph Müller[4], Sibyll Schaphoff[4],*
*Elke Stehfest[2], Detlef P. van Vuuren[2]*
[1] *Wageningen Environmental Research, P.O. Box 47, 6700 AA, Wageningen, the Netherlands*
[2] *PBL Netherlands Environmental Assessment Agency, The Hague, The Netherlands*
[3] *Geoscience & Remote Sensing, Delft University of Technology, Delft, the Netherlands*
[4] *Potsdam Institute for Climate Impact Research, Potsdam, Germany*
[*] *Corresponding author: maarten.braakhekke@gmail.com*

## 12 Abstract

We present an extension of the dynamic global vegetation model LPJmL to simulate planted forests
intended for C sequestration. We implemented three functional types to simulate plantation trees in
temperate, tropical, and boreal climates. The parameters of these functional types were optimized to
fit target growth curves (TGCs). These curves represent the evolution of stemwood C over time in
typical productive plantations and were derived by combining field observations and LPJmL estimates
for equivalent natural forests. While the calibrated model underestimates stemwood C growth rates
compared to the TGCs, it represents substantial improvement over using natural forests to represent
afforestation. Based on a simulation experiment in which we compared global natural forest versus
global forest plantation, we found that forest plantations allow for much larger C uptake rates on the
time scale of 100 years, with a maximum difference of a factor 1.9, around 54 years. In subsequent
simulations for an ambitious but realistic scenario in which 650 Mha (14% of global managed land,
4.5% of global land surface) is converted to forest over 85 years, we found that natural forests take up
37 PgC versus 48 PgC for forest plantations. Comparing these results to estimations of C sequestration
required to achieve the 2°C climate target, we conclude that afforestation can offer a substantial
contribution to climate mitigation. Full evaluation of afforestation as a climate change mitigation
strategy requires an integrated assessment which considers all relevant aspects, including costs,
biodiversity, and trade-offs with other land-use types. Our extended version of LPJmL can contribute
such an assessment by providing improved estimates of C uptake rates by forest plantations.



# 1 Introduction

It is increasingly clear that the stringent climate targets of the Paris Agreement cannot be achieved without negative emissions, i.e. net removal of carbon (C) from the atmosphere, later during the 21st century to compensate for emissions in the first half of the century (Gasser et al., 2015; Rogelj et al., 2018). Of the many proposed techniques to achieve C uptake, the two options currently most discussed for large-scale implementation are bioenergy in combination with carbon capture and storage and afforestation (Williamson, 2016). Both approaches will require considerable amounts of land and thus compete with other land-use functions, for example food production and biodiversity. While bioenergy is receiving considerable attention (van Vuuren et al., 2013), less consideration has been given to afforestation as a tool for land-based mitigation. C uptake occurs when natural vegetation is allowed to grow back on former croplands and pasture. While deliberately taking cropland or pasture out of production may involve costs, the direct management costs of natural regrowth are negligible. The carbon uptake rate of such natural regrowth, however, will usually achieve only a fraction of the potential C uptake rate at short time scales. Considerably higher C uptake rates are possible by planting forests (Paquette and Messier, 2010). Assisting regrowth by planting trees can substantially boost growth rates compared to natural forests because initial stages of primary succession (with herbaceous or shrub vegetation) are skipped and because fast-growing tree species can be selected. Moreover, trees are usually planted as saplings, cultivated under controlled conditions, which improves chance of successful establishment compared to development from seeds (Gladstone and Thomas Ledig, 1990).

Assessing the potential of land-based approaches for climate mitigation requires reliable estimates of C sequestration rates. Process-based models, such as dynamic global vegetation models (DGVMs), are a crucial tool for providing these estimates. DGVMs simulate carbon stocks and fluxes based on mechanistic descriptions of underlying processes, such as photosynthesis and organic matter decomposition in relation to environmental conditions. However, since the focus of DGVM development has traditionally been on natural ecosystems, very few of these models have an explicit representation of planted forests. Therefore, previous modelling studies on large-scale afforestation represented afforestation as natural regrowth (Krause et al., 2017), in some cases applying corrections to account for higher growth rates (Humpenoder et al., 2014; van Minnen et al., 2008).

In this paper we present an updated version of the DGVM LPJmL (Bondeau et al., 2007; Schaphoff et al., 2013), modified to explicitly represent afforestation. Three new plant functional types have been implemented in order to represent planted forests in temperate, tropical, and boreal regions. The parameters of these plantation types were estimated based on observations of stemwood carbon from real-world forest plantations. Using this new LPJmL version we present a global assessment of



potential carbon sequestration rates in forest plantations and compare these to rates achieved by
letting forests grow back naturally.

# 2   Methods

## 2.1 The LPJmL dynamic global vegetation model

LPJmL (Lund-Potsdam-Jena Managed Land) is a global process-based model simulating vegetation
dynamics and fluxes of carbon and water in the vegetation and soil of terrestrial ecosystems (Bondeau
et al., 2007; Schaphoff et al., 2013; Sitch et al., 2003), including agricultural land and biomass
plantations for bioenergy production (Beringer et al., 2011). The model runs primarily on a daily time
step, except for C allocation, vegetation dynamics, and disturbances for natural vegetation and
biomass plantations, which are resolved annually. Forcing consists of monthly climate variables (air
temperature, precipitation, cloud fraction, and number of wet days per month)—which are
interpolated to daily values (Gerten et al., 2004)—and annual atmospheric $CO_2$ concentrations. Using
a combination of plant physiological relations, generalized empirically established functions, and plant
trait parameters, LPJmL simulates processes such as photosynthesis, plant growth, maintenance and
regeneration losses, fire disturbance, soil moisture dynamics, runoff, evapotranspiration, irrigation
and vegetation structure (Schaphoff et al. 2013). Natural vegetation is represented as a number of
plant functional types (PFTs): aggregated vegetation classes representing variation in leaf-type
(broadleaf, needleleaf), phenology (summergreen, evergreen, raingreen), and climate preference
(boreal, temperate, tropical). Most model parameters related to vegetation are defined separately for
each PFT. The model simulates the occurrence of each PFT based on bioclimatic limits and competition
with other PFTs for resources. Agricultural ecosystems are handled in a separate module and are
represented by a range of crop functional types (Bondeau et al., 2007). Additionally, two woody and
one herbaceous PFTs are implemented to simulate short-rotation bioenergy plantations (Beringer et
al., 2011). Area fractions specifying allocation to different land-use types are part of the model input.
Finally, the model can simulate river discharge and surface water reservoirs, and several types of
irrigation. LPJmL has been coupled to the IMAGE integrated assessment model, serving as the land
surface component (Müller et al., 2016; Stehfest et al., 2014).
In all simulations for this study the model was forced by semi-constant monthly climate input,
representative for the period 1980–2010. This dataset was derived by repeating a cycle of detrended
time series for this period, taken from the CRU TS3.23 global gridded (0.5°×0.5° degrees) climate
dataset (Harris et al., 2014). For simplicity we chose to ignore the effect of atmospheric $CO_2$
concentration change at this stage, hence this variable was held fixed at the mean global value for



1980–2010 (362.4 ppmv). Fire disturbance was not considered. Further information on the model
input and configuration is given in subsequent sections.

## 2.2 Forest plantations

LPJmL was extended to represent forest plantations. Specifically, a new land-use type was added, as
well as three functional types to represent plantation trees in temperate, tropical, and boreal
plantations. These types—referred to as forest plantation functional types (FPFTs)—are derived from
the natural PFTs temperate broadleaved summergreen tree, tropical broadleaved evergreen tree, and
boreal needleleaved evergreen tree, respectively. The occurrence of the FPFTs is subject to the same
establishment and mortality rules used for natural PFTs. However, the bioclimatic limits are set such
that they do not overlap, hence co-occurrence of different FPFTs in a single grid cell is rare, occurring
only when climate fluctuates near a boundary between two types..
Structurally, the implementation largely follows that of the woody bioenergy plantations implemented
in LPJmL (Beringer et al., 2011), which in turn are based on equivalent natural PFTs. Contrary to
bioenergy trees, forest plantations are not automatically clear-cut after a fixed rotation period, but a
fraction of the plantation fraction may be harvested, specified as model input. However, for the
purpose of this study, harvest was set to zero. Forest plantation PFTs also differ from other PFTs with
regard to establishment of new trees. A fixed initial planting density ($P_{init}$) was introduced, which
determines the number of trees per unit area at planting. After planting, establishment of new trees
occurs similar to natural PFTs: at fixed maximum rate, downscaled according to an exponentially
declining function of foliar projective cover. Generally, stand density will decrease after plantation due
to self-thinning, implemented according to Reineke's rule (Reineke, 1933), which relates stem
diameter to crown area. When the area-fraction forest plantations in a given grid cell increases over
time, establishment is determined as a combination of $P_{init}$ and the standard establishment rate,
weighted according to the old forest plantation fraction and the fraction added.





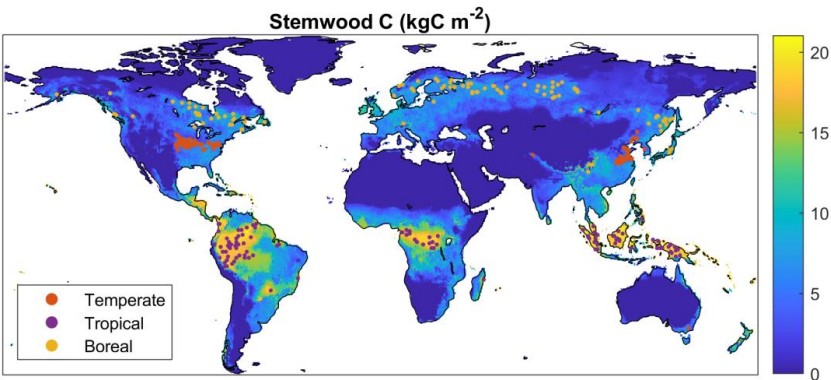

Figure 1. Location of the grid cells included in calibration simulations (100 per FPFT). The map shows simulated stemwood C (kgC m$^{-2}$) for a simulation with potential natural vegetation.

## 2.3 Calibration

### 2.3.1 General setup

To obtain realistic growth rates, we calibrated several FPFT-specific parameters, based on published observed growth data for forest plantations. Ideally, calibration of dynamic vegetation models should be performed using detailed observations for a given site. However, this requires a large amount of data, both for model input and to compare to model output to assess performance. While much data on growth of forest plantations has been published, the number of forest plantation sites for which calibration data as well as data for model input are available for sufficiently long time periods is not enough to derive globally applicable parameter sets. Therefore, we chose a different approach. Rather than aiming to reproduce site-level observations, we calibrated the model in order to obtain desired mean biome-level behavior, for each of the three FPFTs. For every iteration in the calibration, the model was run for a selection of 100 grid cells from the spatial domain of the FPFT being calibrated. Subsequently, model output for the relevant variables were aggregated over all grid cells and compared to observed values to determine model performance.

Within the spatial domain for a given FPFT, many grid cells exist where growth is marginal due unfavorable climate and/or soil properties. The observations used in the calibration are not representative for these locations, since forest plantations from which data have been retrieved can be assumed to represent locations where productivity is sufficient for economic profitability. Therefore, rather than choosing grid cells randomly, the selection was limited to locations for which LPJmL simulates relatively high productivity. This was done based on results from a 300-year simulation with only natural vegetation, in the same setup as used in the calibration (see section 2.3). For each FPFT, 100 cells were selected for which the simulated stemwood C storage of the corresponding natural PFT (see section 2.2) exceeds the 70% percentile over the complete domain where this PFT is



dominant, i.e. has highest foliar projective cover (Fig. 1). During the calibration, LPJmL was run only for
these cells, with land-use type set to forest plantations.
## 2.3.2 Observations
*Target growth curves*
Time series of stand-level stemwood C were collected from various sources in the literature. For the
tropical FPFT we used data from Brown et al. (1986), who derived time series of stemwood biomass
for several species and species groups for tropical forest plantations. Data from natural poplar (populus
x euramericana) forests were taken from Cannell (1982) for the calibration of the temperate FPFT.
Finally, for the boreal FPFT we used data for Scots Pine (Pinus Sylvestris) plantations from Vanninen et
al. (1996). Outliers in the observations were removed using Hampel filtering (Pearson, 2002). The data
are depicted in Figure 2.
Since most forest plantations are grown for timber production, they are harvested approximately at
the optimal rotation length for maximum wood production, which is well before the trees reach
maturity. Hence, growth data for higher tree ages are scarce. Calibrating LPJmL against these
observations alone would result in excessive weight on the earlier part of the curve, leading to
unpredictable results for the later part. Therefore, we did not use the observations directly in the
LPJmL calibration but used them to derive growth curves representing the typical growing behavior of
productive plantations for each FPFT. We refer to these as the target growth curves (TGCs). The general
structure of the TGCs is given by the Chapman-Richards function, which is widely used to model forest
growth (e.g. Von Gadow and Hui, 1999). It defines the stemwood C ($C_{SWC}$) at time $t$ as:

$$C_{SWC}(t) = C_{SWC,max}\left(1 - e^{-k\,t}\right)^{p}, \qquad (1)$$

where $C_{SWC,max}$ is the maximum $C_{SWC}$, $k$ is the growth rate, and $p$ is a shape parameter determining
the timing of maximum growth. These parameters were estimated using a Bayesian non-linear
regression approach. The scarcity of high-age observations was solved by constraining $C_{SWC,max}$ using
a prior distribution based on LPJmL output for the corresponding natural PFTs, from the 300-year
simulation used to select the calibration grid cells (see section 2.3.1). Specifically, for each FPFT we
used the mean simulated stemwood C of the last 10 simulation years, averaged over the 100
calibration cells as a representative value.
The parameters were estimated using MCMC sampling. The sample with highest posterior density,
together with the variances over the marginal posterior distributions, were used in the LPJmL
calibration. Further details are given in the supplemental text.



# 1  *Additional constraints*

Initial tests showed that parameter sets derived by calibration with the TGCs alone result in
unrealistically high values of net primary production (NPP), leading to similarly high litter fluxes and
soil carbon storage. This was traced to a higher carbon use efficiency (CUE)—the ratio of NPP to gross
primary productivity—and a lower vegetation carbon turnover time ($\tau_{vegC}$; vegetation C to NPP ratio)
compared to the natural PFT counterparts. Therefore, to assure realistic carbon fluxes and storage, we
implemented additional constraints for these variables in the calibration. These constraints are based
on LPJmL simulations for the natural PFT counterparts of the FPFTs, similar to the maximum stemwood
C of the target growth curves (see Table 2 and Figure 4).
Additionally, it was found that certain parameter sets, while leading to acceptable mean results, cause
simulated trees for certain cells to die-off repeatedly at regular intervals. In order to avoid this we
modified the calibration such that a penalty was added to the cost function when this occurs.

Table 1. LPJmL parameters included in the calibration. Prior mode refers to the most probable value indicated by the prior distribution.

| Parameter | Description | Units | Temperate | | Tropical | | Boreal | |
|---|---|---|---|---|---|---|---|---|
| | | | Prior mode | Estimate | Prior mode | Estimate | Prior mode | Estimate |
| $\alpha_a$ | Fraction of PAR assimilated at ecosystem level, relative to leaf level | - | 0.5 | 0.61 | 0.5 | 0.60 | 0.5 | 0.53 |
| $g_{min}$ | Minimum canopy conductance | mm s$^{-1}$ | 0.5 | 0.62 | 0.5 | 0.48 | 0.3 | 0.28 |
| $E_{max}$ | Maximum transpiration rate | mm d$^{-1}$ | 5 | 5.5 | 7 | 11.7 | 5 | 6.07 |
| $r$ | Maintenance respiration coefficient | gC gN$^{-1}$ d$^{-1}$ | 1.2 | 1.5 | 0.2 | 0.16 | 1.2 | 1.2 |
| $k_{allom1}$ | Allometry parameter 1; relates crown area to stem diameter | - | 100 | 126 | 100 | 166 | 110 | 86.4 |
| $k_{allom2}$ | Allometry parameter 2; relates tree height to stem diameter | - | 40 | 53.9 | 40 | 41.1 | 40 | 36.5 |
| $k_{allom3}$ | Allometry parameter 3; relates tree height to stem diameter | - | 0.67 | 1.02 | 0.67 | 0.84 | 0.67 | 1.2 |
| $lr_{max}$ | Leaf to root ratio under non-water stressed conditions | - | 1 | 1.2 | 1 | 1.5 | 1 | 1.5 |
| $C_{sapwood;sapl}$ | Sapwood C of saplings | gC m$^{-2}$ | 1.2 | 1.02 | 1.2 | 1.2 | 1.2 | 1.4 |
| $LAI_{sapl}$ | Leaf area index of saplings | - | 1.5 | 1.4 | 1.5 | 1.4 | 1.5 | 1.7 |
| $\alpha_{leaf}$ | Leaf longevity | months | 0.5 | 0.46 | 2 | 1.4 | 4 | 4.0 |
| $\tau_{sapwood}$ | Turnover time of sapwood | yr | 20 | 20 | 20 | 46.3 | 20 | 15.7 |
| $\tau_{leaf,root}$ | Turnover time of leaves and roots | yr | 1 | 1.3 | 2 | 1.8 | 4 | 4.7 |
| $P_{init}$ | Planting density | m$^{-2}$ | 0.15 | 0.15 | 0.15 | 0.15 | 0.15 | 0.17 |
| $k_{mort1}$ | Maximum mortality rate | yr$^{-1}$ | 0.03 | 0.064 | 0.03 | 0.058 | 0.03 | 0.048 |





### 2.3.3 Parameter estimation

In the calibration 15 parameters were estimated, separately for each FPFT (Table 1). The calibration was performed on a transformed scale (logit for $\alpha_a$; log for all other parameters), in view of the lower bound at zero (and upper bound at 1 for $\alpha_a$). We applied a Bayesian cost function, including informative prior distributions. Priors express belief about reasonable parameter values before the calibration in the form of probability distributions and help to avoid unrealistic values, particularly for parameters that have little influence on the relevant model output. The priors were chosen such that their central tendency reflects existing parameter values for the corresponding natural PFTs, with a relatively wide variance to avoid overly strong influence on the calibration. Full specification of the priors is given in supplemental text S1.

Similar to the parameters, all observations were transformed in the calibration (logit for CUE; log for all other observations). For the calibration simulations, LPJmL was started from zero vegetation and soil C and run for a period of 300 years, sufficient for the vegetation C to reach equilibrium with reasonable parameter values. LPJmL simulates heartwood and sapwood C pools, but does not distinguish between stem, branches, and coarse roots. For the purpose of the calibration, we assumed that all heartwood and 66% of the sapwood is located aboveground (Müller et al., 2016), and 84% of aboveground wood is located in the stem (which is representative for mature trees (Pretzsch, 2010).

After simulation, the Chapman-Richards function was fitted to the time-series of simulated stemwood C for the 100 grid cells (using non-linear least squares) to derive FPFT-mean estimates of $C_{\text{SWC,max}}$, $k$, and $p$ based on LPJmL predictions. Carbon use efficiency and vegetation turnover time were determined for the last 10 years of the simulation, averaged over the 100 grid cells. The observations were subsequently compared to the corresponding observations to determine log-likelihood, and combined with log prior density to determine the overall cost $C(\theta)$ for the given parameter set $\theta$. Further details are given in supplemental text S2.

Table 2. Observations and corresponding fits for the 100 included grid cells included in the calibration. Observed values correspond to the mode of the likelihood function.

| Symbol | Description | Units | Temperate | | Tropical | | Boreal | |
|---|---|---|---|---|---|---|---|---|
| | | | Obs. | Fit | Obs. | Fit | Obs. | Fit |
| $C_{\text{SWC,max}}$ | Growth curve parameter; maximum stemwood C | kgC m⁻² | 6.77 | 6.45 | 15.62 | 15.74 | 7.45 | 7.56 |
| $k$ | Parameter of growth curve | yr⁻¹ | 0.197 | 0.0420 | 0.0566 | 0.0301 | 0.0500 | 0.0257 |
| $p$ | Parameter of growth curve | - | 3.91 | 3.37 | 1.59 | 1.69 | 4.28 | 4.64 |
| CUE | Carbon use efficiency; NPP to GPP ratio | - | 0.380 | 0.342 | 0.458 | 0.448 | 0.460 | 0.427 |
| $\tau_{\text{vegC}}$ | Vegetation C turnover time; Vegetation C to NPP ratio | yr | 16.86 | 18.37 | 21.92 | 17.99 | 22.27 | 19.52 |



The optimal parameter set (with minimal value of $C$) was derived using the genoud algorithm (Mebane
Jr. and Sekhon, 2011) which combines a genetic algorithm with a gradient search approach. This
algorithm has previously been applied to calibrate LPJmL (Forkel et al., 2014). Additional description is
given in supplemental text S2.

## 2.4 Global simulations

After calibration, several global simulations were performed. First, in order to assess sequestration
potential of afforestation, a simulation was run in the same setup as used for the calibration, i.e.
starting with zero vegetation and soil C and with land fully allocated to forest plantations and running
for 300 years so that vegetation C pool can reach equilibrium. Additionally, a simulation with land fully
allocated to natural vegetation was performed, to compare natural regrowth and afforestation as land-
based mitigation options.
Second, we applied the model for an ambitious scenario of large-scale afforestation, assuming that
from 2015 onwards approximately 14% of global managed land is (corresponding to 650 Mha or 4.5%
of global land surface) gradually replaced by forest plantations over the course of 85 years. This
afforestation area is in line with the average land area used for land-based mitigation (both bio-energy
and afforestation) in 1.5 degree mitigation scenarios in Integrated Assessment Models (Doelman et
al., in review; Rogelj et al., 2018). To bring soil C to reasonable values, the simulation was initialized by
two spin-up phases: 1) a 1000-year phase with natural vegetation only until 1901, and 2) a phase from
1900 to 2015 with transient cropland and pasture fractions, based on the HYDE dataset (Klein
Goldewijk et al., 2010). From 2015 forest plantation area was increased and crop and pasture area
was, balancing each other so that total area of managed (i.e. non-natural) land remained constant.
From 2100 the simulation was continued for another 50 years with constant land-use. For this analysis,
two complementary simulations were performed. First, a simulation where fractions of natural
vegetation were increased, instead of forest plantations, and second, a "baseline" simulation where
land-use fractions were held constant in time from 2015. Supplemental Figures S2 and S3 depict the
development of land-use fractions for the three scenarios.

## 3 Results

## 3.1 Target growth curves

Figure 2 depicts the stemwood C observations for the three FPFTs, LPJmL simulations for the
corresponding natural PFTs, and the target growth curves (TGCs) resulting from the fitting procedure.
The values of the maximum stemwood C ($C_{\mathrm{SWC,max}}$), growth rate ($k$), and shape parameter ($p$) and
their marginal variances are given in Table 2 (see also Figure S1). As expected, the observations show
substantially higher growth rates than simulated for the natural PFTs, as well as an earlier timing of



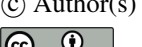

maximum growth. The TGCs represent a compromise between the observations and the $C_{\text{SWC,max}}$ for
natural PFTs, predicted by LPJmL: the initial high growth rate is representative for the observations,
while $C_{\text{SWC,max}}$ is closer to that of the simulated natural PFTs and notably lower than the level
indicated by the observations.
The tropical FPFT has substantially higher $C_{\text{SWC,max}}$, approximately twice as high as the other two
FPFTs. With respect to the relative growth rate $k$, however, the tropical FPFT is comparable to the
boreal FPFT. The temperate FPFT approaches its maximum stemwood approximately four times faster.
The boreal TGC has the highest value of $p$, resulting in a later timing of maximum growth.

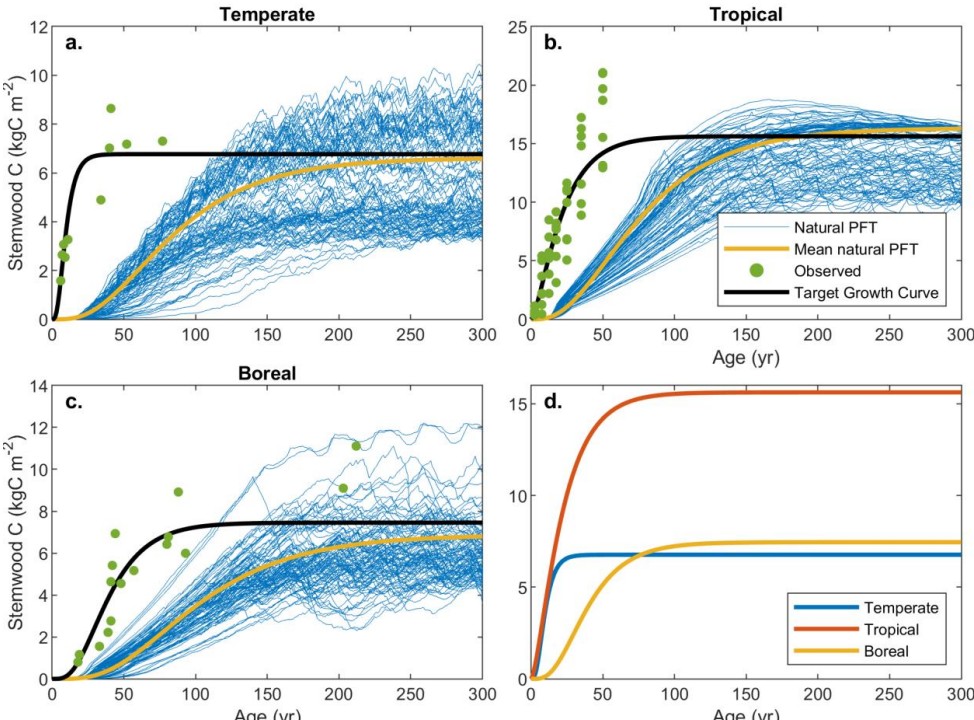

Figure 2. Target growth curves for stemwood C, associated observations and LPJmL output for natural vegetation in the cells selected for calibration.



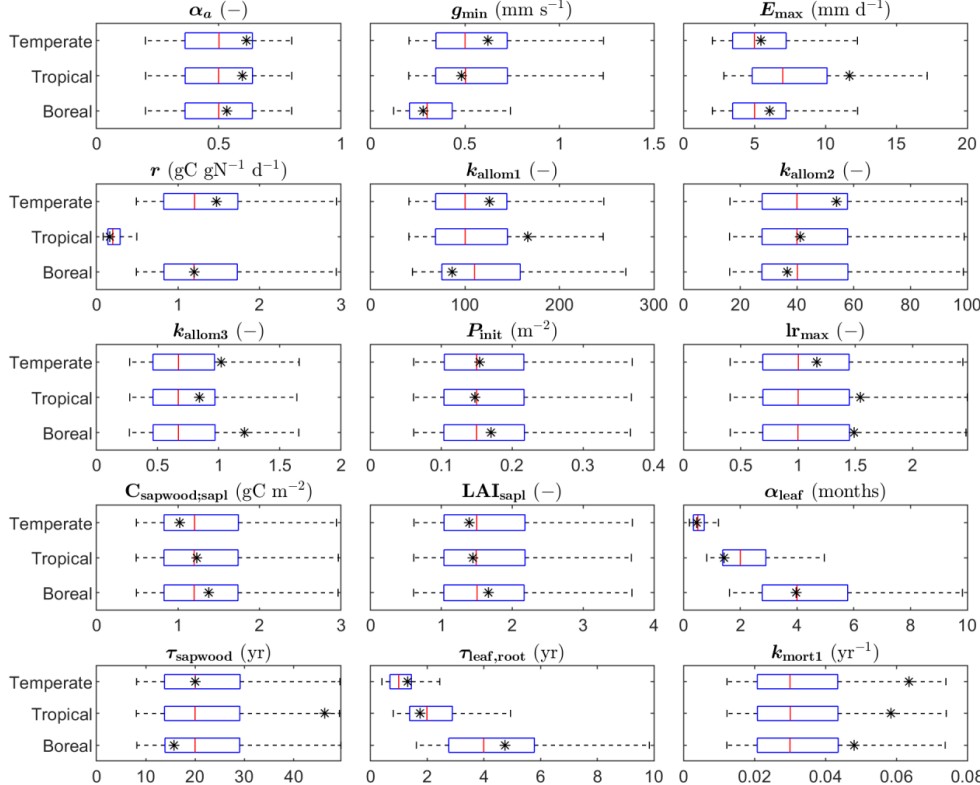

Figure 3. Prior distributions and estimated final values of the FPFT parameters estimated in the calibration. The boxplots indicate the 5% and 95% percentiles (whiskers) the median (red line) and 25% and 75% percentiles (box) or the priors. The final parameter estimate is indicated by the asterisk ($*$). See Table 1 for explanation of the parameters.

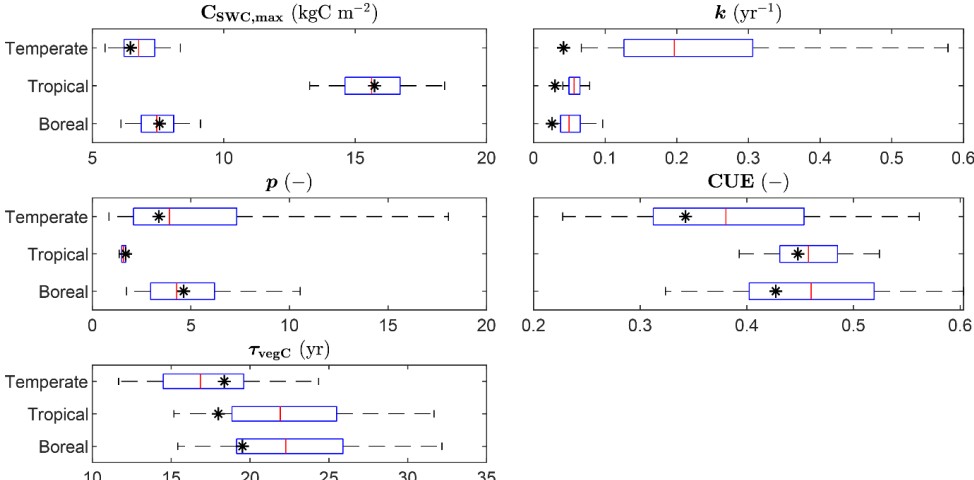

Figure 4. Ranges of the observations used in the calibration and LPJmL estimates after calibration. The boxplots indicate the 5% and 95% percentiles (whiskers) the median (red line) and 25% and 75% percentiles (box) of the likelihood function. The final fitted value is indicated by the asterisk ($*$). $C_{\mathrm{SWC,max}}$: maximum stemwood C, $k$: growth rate; $p$: shape factor, $\mathrm{CUE}$: carbon use efficiency (NPP to GPP ratio); $\tau_{\mathrm{vegC}}$:vegetation C turnover time (vegetation C to NPP ratio).





## 3.2 LPJmL calibration

The parameter estimates resulting from the calibration are shown in Figure 3, together with the range
of the prior distributions. Most estimates are within interquartile range of the priors, but for several
parameters the calibration resulted in relatively strong changes, in particular $k_{\mathrm{mort1}}$, which controls
mortality due to low growth efficiency. Specifically for the tropical FPFT, the estimates also clearly
deviate from the prior for $E_{\mathrm{max}}$, $k_{\mathrm{allom1}}$, $\mathrm{lr}_{\mathrm{max}}$, and $\tau_{\mathrm{sapwood}}$.
The ranges of the observed variables are depicted in Figure 4, together with the LPJmL predictions for
the calibration grid cells, based on the optimized parameter sets. The parameter $C_{\mathrm{SWC,max}}$, is fit well
by the model for all FPFTs. However, $k$ is clearly underestimated for all three FPFTs, compared to the
observed ranges. This affects the simulated growth curve for stemwood C, as shown in Figure 5. In the
LPJmL simulation, vegetation needs a longer time to reach its maximum stemwood biomass than the
target growth curve. Nevertheless, the growth rate based on the optimized parameters represents a
substantial improvement compared to the natural PFT counterparts. The carbon use efficiency (CUE)
and vegetation turnover time ($\tau_{\mathrm{vegC}}$) are also reasonably well fitted.

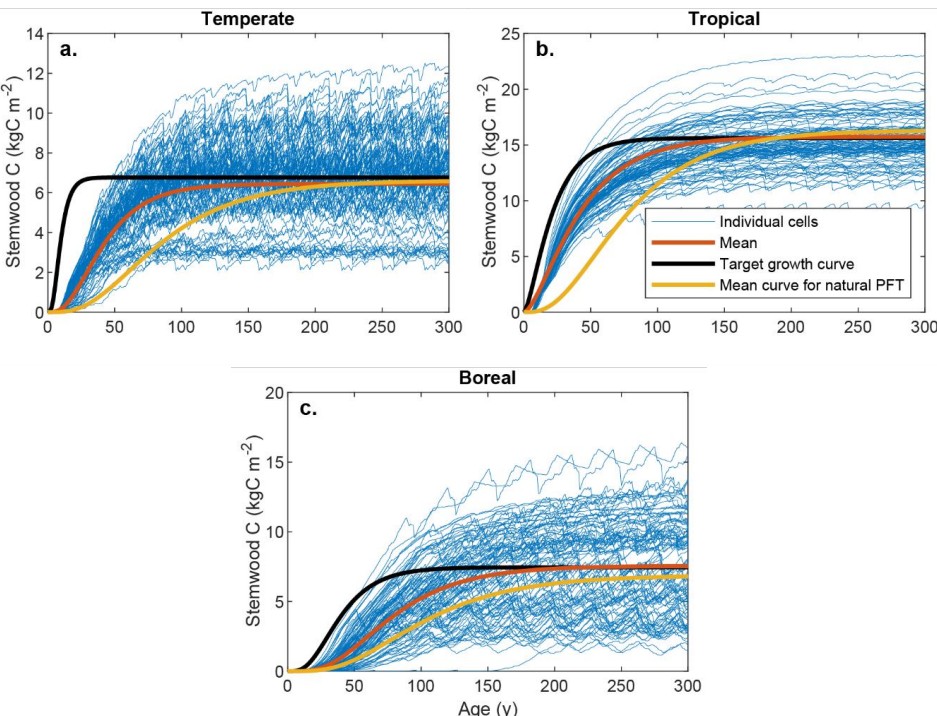

Figure 5. Predicted stemwood C for 100 calibration grid cells of each FPFT based on the optimal parameter sets. Note the different scales of the y axes.



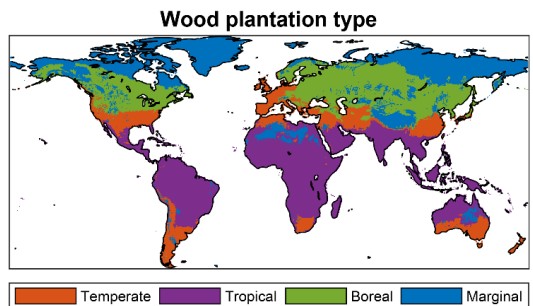

Figure 6. Spatial distribution of the different forest plantation functional types resulting from the bioclimatic limits. In marginal regions no trees are simulated, but grass may be present.

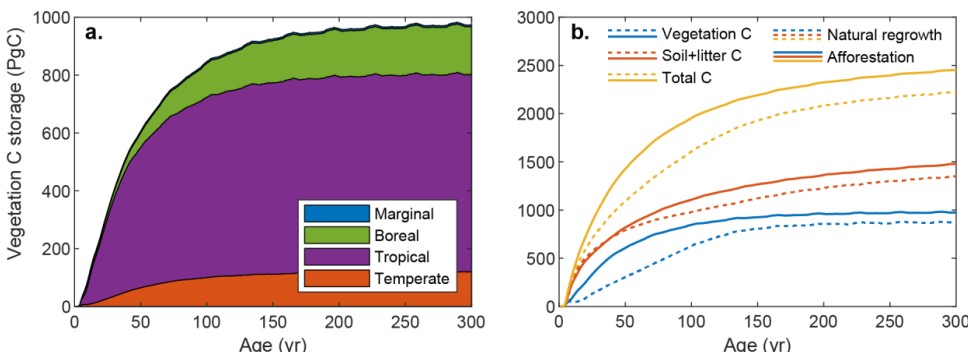

Figure 7. Global total ecosystem C over time for simulations with global forest plantations or global natural vegetation. (a) vegetation C storage per biome for the global forest plantation simulation only; (b) C storage per compartment for both simulations.

## 3.3 Global simulations

### 3.3.1 Global afforestation and natural regrowth

Figure 6 depicts the predicted spatial distribution of forest plantation functional types for a global

simulation experiment with land fully allocated to forest plantations. The total area for the temperate,

tropical and boreal plantation types is 2,472 Mha ($10^{10}$ m$^2$), 6,242 Mha, and 3,094 Mha, respectively,

corresponding to 17%, 43%, and 22%, of global land surface. In 2,579 Mha (18%) no tree growth is

simulated due to a too cold or too dry climate. Note that in many non-marginal regions tree growth

may still be very low due to unfavorable conditions—the depicted distribution simply results from the

bioclimatic limits set in the model. The distribution the FPFTs corresponds roughly to zones C, A, and

D of the first level of the Köppen climate classification (Peel et al., 2007). Since there is no type for

forest plantations in arid climates, the three FPFTs extend also into desert regions.

The global vegetation C stock over time is depicted in Figure 7a (cf also Figure S2). Tropical plantations

contribute most to C storage due to their larger area and higher productivity. Comparison with the

simulation where all land is allocated to natural vegetation shows considerably faster C uptake for

forest plantations (Figure 7b), with a maximum difference of 308 PgC (193%), after 54 years. After 300

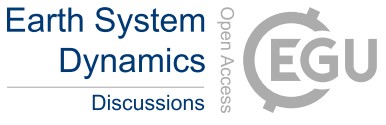

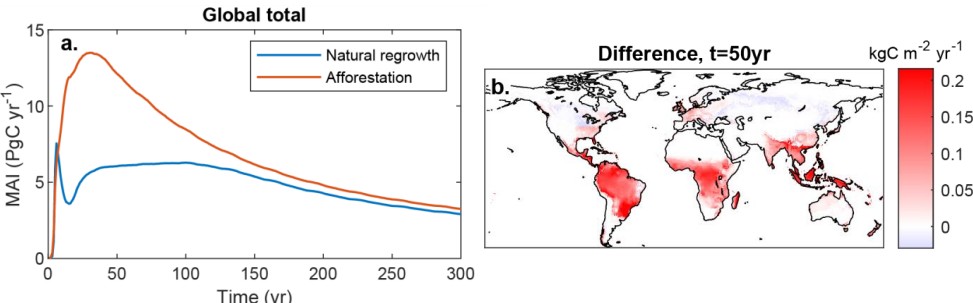

Figure 8. Mean ecosystem sequestration rate (Mean annual increment, MAI), determined as total C storage divided by time since start for LPJmL simulations with only forest plantations or only natural vegetation. (a) global total versus time; (b) difference between (afforestation minus natural regrowth) after 50 years.

years, global vegetation C is 102 PgC higher (112%) for afforestation simulation. Soil and litter C storage
is also proportionally higher for forest plantations. Note that the soil and litter C uptake rate is
extremely high due to the fact that the simulation was started with zero C. In reality soil C will already
be present before land-use change and uptake will be much slower, possibly even negative, depending
on previous land-use.
The potential for C uptake is illustrated by the mean annual increment (MAI) of vegetation C since the
start of plantation (Figure 8). There are remarkable differences between the two simulations. After an
initial similar increase, MAI sharply drops after approximately 10 years for natural regrowth, while for
afforestation MAI keeps rising until approximately 30 years. The behavior for natural regrowth can be
explained by vegetation succession, leading to a shift from grasses to trees. This succession does not
occur for forest plantations, where trees start growing immediately, resulting in a substantially higher
MAI in the early part of the simulation. From spatial differences in MAI after 50 years it is evident that
tropical regions contribute most to this difference.
### 3.3.2 Transient afforestation and natural regrowth
Figure 9 depicts results of the global simulation scenarios with gradual increase in forest, applying
either afforestation or  natural regrowth. Since changes in C storage—particularly for the soil—result
also from land-use changes before



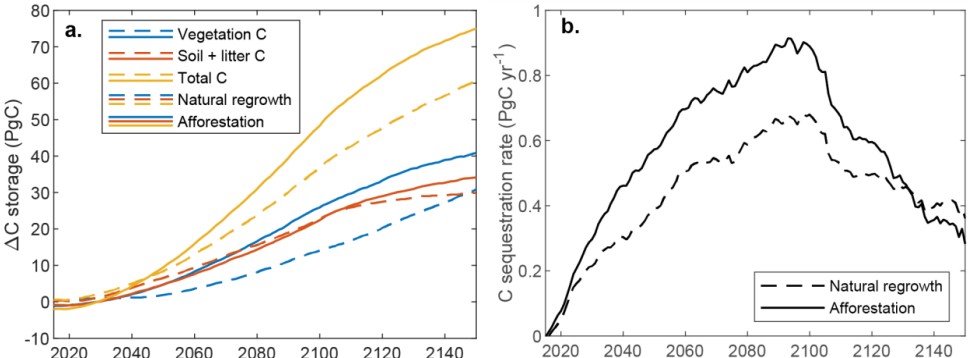

Figure 9 Results of the simulations for gradual afforestation and natural regrowth. Both graphs show differences relative to the baseline simulation with constant land-use. (a) global C storage; (b) global C sequestration rate, smoothed using a 10-year moving average window.

2015, we focus on the difference in global C stocks compared to the baseline simulation with constant
land-use from 2015. Until 2015 all three simulations have very similar results, but small differences
arise from the stochastic generation of daily precipitation. Gradual afforestation of 650 Mha land
between 2015 and 2100 results in 19, 48, and 75 PgC additional C storage by 2065, 2100, and 2150,
respectively, versus 16, 37, and 61 PgC for natural regrowth. Most of the difference between the two
simulations is due to vegetation C, but from 2100 the difference for soil C grows and would ultimately
dominate, had the simulation been continued after 2150. Global C sequestration rate peaks between
2090 and 2100 at approximately 0.91 and 0.68 PgC yr$^{-1}$ for afforestation and natural regrowth, with
average rates of 0.25 and 0.19 PgC yr$^{-1}$ until 2100. Around 2130 global sequestration rates are higher
for the natural regrowth simulation because land-use remains constant from 2100, allowing natural
ecosystems to "catch up".
# 4   Discussion
## 4.1 LPJmL calibration
### 4.1.1 Parameters changes
Compared to the prior distributions—which are largely based on values for corresponding natural
PFTs—the calibration resulted in a substantial shift for several parameters. We will discuss the more
notable changes. First, $k_{\mathrm{mort1}}$ is substantially higher compared to the prior mode for all FPFTs. This
parameter controls tree mortality related to low growth efficiency, which is defined as the ratio of the
annual net biomass increment to leaf area. A high value of $k_{\mathrm{mort1}}$ results in higher mortality under
unfavorable conditions. The increase of this parameter can be explained by the fact that the target
growth curves have substantially higher growth rates than the natural PFT equivalents, while maximum
biomass is approximately the same. The apparent conflict between these two constraints can in part
be resolved by increasing first-order mortality. A higher value for $k_{\mathrm{mort1}}$ for forest plantation trees is
not necessarily unrealistic since it is likely that fast-growing tree species have low tolerance for low
growth conditions (Pacala et al., 1996).
The parameter $k_{\mathrm{allom3}}$, which relates tree stem diameter to tree height, has also substantially
increased for all FPFTs. Higher values of this parameter mean higher trees for the same diameter,
resulting in higher maximum biomass per tree. Again, this is in agreement with field observations,
which have shown a positive relationship between tree growth rate and this parameter (Martinez Cano
et al., 2018)
The maximum leaf-to-root mass ratio, $\mathrm{lr}_{\mathrm{max}}$, is also high compared to the prior, particularly for the
tropical and boreal FPFTs. This causes higher allocation of C to leaves, compared to roots, which
positively affects growth rate via leaf area index and absorbed photosynthetically active radiation.
Conversely, in LPJmL, lower root biomass does not reduce growth since there is no link between root
biomass and water uptake. Hence, higher values of $\mathrm{lr}_{\mathrm{max}}$ unequivocally lead to higher productivity in
the model.
For the tropical FPFT, $E_{\mathrm{max}}$, $\tau_{\mathrm{sapwood}}$, and $k_{\mathrm{allom1}}$ have increased. $E_{\mathrm{max}}$ is the maximum water
transport capacity and controls the transpiration rate. $\tau_{\mathrm{sapwood}}$ is the turnover time of sapwood—
higher values result in more sapwood biomass, which allows for larger leaf area. Finally, $k_{\mathrm{allom1}}$ relates
crown area to stem diameter. The generally stronger shifts for the tropical FPFT compared to
temperate and boreal is explained by the lower uncertainties of parameters $k$ and $p$ of the target
growth curve (Figure 4, S1), which is in turn caused by the larger number of stemwood C observations
(Figure 2).

## 4.1.2 Fit to observations

The calibration resulted in good fits with respect to most observations, with the exception of the
growth rate parameter of the target growth curves. Despite substantial improvement compared to the
corresponding natural PFTs, this parameter is underestimated for all three FPFTs. As a result, predicted
initial C uptake rates are lower than implied by the stemwood C observations, possibly underestimating
the potential efficacy of forest plantations for climate mitigation.
As discussed in section 2.3.2, we incorporated data into the calibration to constrain NPP to GPP ratio
and vegetation C turnover time to values similar to that of the corresponding natural PFTs. Earlier
calibrations, in which these constraints were not included, yielded a substantially better fit to the
growth rate but with unrealistic litter fluxes, which points to a trade-off between the fit to these
observations. From a mass-balance perspective this result is explicable: fast growth requires high NPP,
which will result in high litter fluxes once vegetation reaches equilibrium biomass. This is exacerbated



by the fact that we constrained maximum stemwood C ($C_{\mathrm{SWC,max}}$) at levels close to that of the
corresponding natural PFTs. The fact that fast growth results in very high litter fluxes when trees reach
equilibrium relates to the fact that LPJmL does not represent certain mechanisms that lead to declining
productivity with age (Zaehle et al., 2006). In reality, NPP reduction with age has been frequently
observed (Ryan et al., 1997). Multiple reasons for this phenomenon have been proposed, but the
leading hypothesis is that hydraulic resistance increases with tree height due to longer distance
between soil and leaves (Ryan and Yoder, 1997). This result in a lower photosynthesis rates and gross
productivity (GPP). Since LPJmL does not include such mechanisms, it is mostly representative for
mature forests. Incorporation of a more realistic representation of age-dependence of forest growth
rate is likely to improve fit to observations (Zaehle et al., 2006).

## 4.2 Global simulations

### 4.2.1 Global afforestation versus natural regrowth

Despite the underestimates growth rates, our results show C can be sequestered substantially faster
by forest plantations compared to natural regrowth (Figure 7), particularly in the first 50 years
following land conversion. The largest potential for plantations lies in tropical regions, which is not
surprising, given that the maximum biomass of tropical FPFTs is more than twice as high compared to
the temperate and boreal FPFTs. In addition to faster C sequestration, LPJmL also predicts a 12% higher
equilibrium global vegetation C pool for forest plantations, despite the fact that the FPFTs were
calibrated to produce a value of $C_{\mathrm{SWC,max}}$ comparable to that of natural equivalent PFTs (Figure 5).
This contradiction is in part explained by a larger productivity of forest plantations in less productive
regions, which were not included in the calibration grid cells. Another reason is a larger spatial extent
of the tropical FPFT compared to that of the natural tropical PFTs.
Combined soil and litter C is also higher for forest plantations after 300 years (Figure 7b), but its
proportion to total ecosystem C (60%) is globally almost identical to that of the natural vegetation,
owing to the constraints on NPP to GPP ratio and vegetation C turnover time included in the calibration
(see section 2.3.2). It is difficult to compare these results to observations for real-world plantations
since studies on this topic have generally compared natural forests to tree plantations for production
of wood or other products, where the effects of harvest and other management on soil C are likely
considerable (Guo and Gifford, 2002; van Straaten et al., 2015). Such effects are not relevant for
plantations intended for C sequestration.

### 4.2.2 Gradual afforestation versus natural regrowth

According to our projections, gradual conversion of 650 Mha managed land to natural forest between
2015 and 2100 results in additional C uptake of 16 and 37 PgC by 2065 and 2100, respectively. If these
lands are converted to forest plantations, the estimated C uptake is 19 and 48 PgC, i.e. 19% and 30%



higher. These should be seen as a conservative estimates, in view of the underestimated growth rates
resulting from the calibration. To put these numbers into perspective, we compare them to results of
Gasser et al. (2015), who estimated the negative emissions needed to limit global warming to 2°C for
a range of scenarios in which both the start time and the rate of reduction of greenhouse gasses varied.
In their most favorable scenario (energy and industry emission reduction starting in 2015 at a rate of
5% per year) they estimated an average cumulative negative emission of 25–100 PgC is needed by
2100, compared to 450–800 PgC in the most unfavorable scenario (energy and industry emission
reduction starting in 2030 at 1% per year). Hence, large-scale forest plantations can offer a substantial
contribution to climate mitigation but will likely not be sufficient.

## 4.3 Comparison to previous work

The results of the simulations for transient afforestation and natural regrowth compare well to results
of previous studies on potential C sequestration rates of forest plantations and natural regrowth. For
example, using the IMAGE integrated assessment model, van Minnen et al. (2008) performed a
simulation experiment based on the IPCC SRES A1B scenario where 831 Mha agricultural land is
converted to permanent forest plantations between 2000 and 2100, taking into consideration land
demand for food production and other uses. They estimated an additional 93 PgC can be sequestered,
but mostly after 2050, when land becomes gradually available due to decreasing population and
increasing agricultural efficiency.
Humpenöder et al. (2014) presented a much more ambitious afforestation scenario, in which 2773
Mha land is converted to forest plantations. The authors used maximum C storage for natural
vegetation predicted by LPJmL, but corrected sequestration rates using stylized growth curves for
plantations in different climate regions. They estimate an additional C uptake of 192 PgC after 80 years.
Roughly converting our estimate to the same land area yields a similar result (205 PgC). This similarity
is not surprising, given that we used the same model, and our FPFTs were calibrated to produce the
same maximum biomass as the natural PFT equivalents.
Potential sequestration rates by natural regrowth were studied by Krause et al. (2017), using the
dynamic global vegetation model LPJ-GUESS. In two scenarios, derived by IMAGE and the agricultural
land-use model MAgPIE, 1119 and 914 Mha were converted to natural lands, resulting in a predicted
additional C uptake of 76 and 55 PgC, respectively, between 2000 and 2099. This compares well with
our estimates for natural regrowth.

## 4.4 Model limitations

In our implementation of planted forests the diversity of plantation tree species is reduced to three
functional types with fixed properties. While the functional diversity of plantation tree species is not



as vast as that of natural forests—especially in the context of C sequestration—the predictions would
likely improve from implementation of additional FPFTs, particularly for the tropical biome. The model
currently predicts a relatively large C storage for dry tropical zone compared to natural regrowth,
which may not be fully realistic, given water limitations. Addition of a dry tropical FPFT would allow for
a more accurate assessment of C sequestration in these regions.
This study does not consider the effects of climate change and $CO_2$ concentration on productivity of
forest plantations. Although there is still considerable uncertainty regarding this topic, accounting for
$CO_2$ fertilization will likely increase the C sequestration potential (Schimel et al., 2015) in both natural
and managed forests. However, in order to properly assess this, it is important to take into account
nutrient limitation to productivity as well (Norby et al., 2010).
We also did not consider possible management options that may improve C uptake rates. In particular,
regular thinning can result in substantially higher C uptake rates (van Minnen et al., 2008). The model
supports harvesting, but this feature was not used in this study. However, continual thinning would
result in export of nutrients from the ecosystem which would ultimately slow down growth rates,
unless plantations are fertilized. Thus, representing regular harvest in LPJmL would also require
representation of nutrient limitation.

## 4.5 Considerations beyond C uptake

Evaluation of afforestation and natural regrowth as strategies for climate change mitigation involves a
range of considerations other than carbon sequestration. First, converting agricultural land to forest
involves a number of costs. For both natural and planted forests this includes price for acquiring land,
while specifically for the latter costs related to establishing and maintaining the plantation are relevant
(e.g. land preparation, planting of seedlings). The costs per unit C sequestered will rise with increasing
area of (planted) forest, mainly due to competition for land (Doelman et al., in review).
Second, the positive effects of carbon uptake of changing land-cover to forest, can be offset due
biophysical changes in the surface energy budget, related to changes in albedo, evapotranspiration,
and surface roughness (Perugini et al., 2017). This may result in a net warming effect, regionally, and
possibly globally, depending on the extent of land-cover change.
Third, the reduction of cropland and pasture might also have a negative impact on food security due
to increased competition for land (Hasegawa et al., 2018). In order to maintain food production for the
growing population, strong intensification of the agricultural sector would be required. Locally, this
will result in a range of negative effects on the environment, due to higher application of fertilizers and
plant protection products, as well as water extraction for irrigation (Smith et al., 2013). Furthermore,



in terms of climate change mitigation, agricultural intensification will likely partially offset the benefits
of afforestation and regrowth, e.g. due to higher $N_2O$ emissions from fertilizers (Burney et al., 2010).
Finally, biodiversity is a particularly important aspect to consider, given that plantation forests have
usually substantially lower species richness than primary or secondary forests (Barlow et al., 2007). A
more balanced solution may be a compromise between biodiversity and C sequestration by
establishing a mixture of native and plantation species, or plantation forest with a native undergrowth
(Barlow et al., 2007; Bremer and Farley, 2010).

# 5   Conclusions

To our knowledge, the extension of LPJmL presented here represents the first model of forest
plantations for C sequestration as part of a DGVM for global-scale applications. Although calibration
of the model still resulted in underestimated growth rates compared to observations of stemwood C,
this represents an improvement over previous approaches. According to our simulations, conversion
of 650 Mha of land to forest over 85 years results in an additional C uptake of 48 PgC for forest
plantation, versus 37 PgC for natural regrowth, with greatest potential in the tropics. We conclude that
large scale afforestation can offer a substantial contribution to C uptake, particularly in a time scale of
approximately 50–100 years. Evaluating afforestation as a strategy for climate change mitigation
requires consideration of all relevant aspects in a comprehensive assessment. Our model can
contribute to such an evaluation by providing improved estimates of C uptake rates.

# Acknowledgements

We gratefully acknowledge Matthias Forkel for advice on using the Genoud algorithm to calibrate
LPJmL.





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
