# Peer review of "Modelling forest plantations for carbon uptake"

_Earth System Dynamics, 2019_

## Referee Comment (RC1) · Anonymous Referee #1 · 27 Apr 2019

The study elaborates on the need for modelling forest plantations for carbon uptake with the LPJmL dynamic global vegetation model, in the context of Paris Agreement and other mitigation goals and targets.

The modification of the PFTs to suit plantation types is a very important contribution. Although, currently, only three broad plantation PFTs have been included, the authors clearly state the limitation of doing so, particularly for tropical systems.

The study also has comprehensively outlined the barriers and limitations of the assumption to promote plantations in all areas and in croplands.

Overall, this is a well written paper, that is comprehensive as well as clear and ad-

dresses a topic of relevance.

---

## Referee Comment (RC2) · Jagmohan Sharma (Referee) · 12 May 2019

Extension of LPJmL DGVM to assess the C sequestration potential of forest plantations is a useful development that helps in considering forest plantation as a global warming mitigation option. Parameterization of forests is challenging, as forest systems are complex entities and there exist vital information/knowledge gaps. The present effort to model the global forests under three broad categories (tropical, temperate and boreal forests) and not including certain factors (natural hazards, drought, fire disturbance, biomass removals etc.) simplifies such complexity. The effort to make use of field observations and comparison with equivalent natural forest to calibrate the model with

regard to biomass productivity has enhanced the value of the study results. The extended model is able to simulate observation-consistent results like plantations showing higher growth rates at least during the first 50 years. Despite major limitations such as only 3 species PFT, one each for tropical, temperate and boreal forests selected to simulate carbon uptake, harvest set to zero and omitting $CO_2$ fertilization, the study still presents an opportunity to advance our capability to assess forestry-based options to mitigate climate change. The limitations quoted by the study bring focus on the areas for future advancement; most important among which is the characterization of PFTs for more number of common plantation species from the three zones. This study reports substantial work in the field of modeling for assessing attributes of forest plantation such as biomass (Carbon) accumulation. I find the methodology appropriate and results useful for publication. Study is likely to encourage more research work pertaining to forests as land use in general and plantations in particular. Authors may however consider revising the zone and carbon density color-coding in figure-1 for easy comprehension. Supplementary information may include methodology steps pertaining to selection of the representative species for the three zones and details of the location and source of observation data Recommended for publication.
* * *

---

## Author Comment (AC1) · 29 Jun 2019

We very much thank the reviewer for this commendation.
* * *

---

## Author Comment (AC2) · 29 Jun 2019

**Reply to comments by Jagmohan Sharma, on "Modelling forest plantations for carbon uptake with the LPJmL dynamic global vegetation model"**

*Maarten C. Braakhekke, Jonathan C. Doelman, Peter Baas, Christoph Müller, Sibyll Schaphoff, Elke Stehfest, Detlef P. van Vuuren*

*The reviewers' comments are included below. Our reply is set in red italic font.*

Extension of LPJmL DGVM to assess the C sequestration potential of forest plantations is a useful development that helps in considering forest plantation as a global warming mitigation option. Parameterization of forests is challenging, as forest systems are complex entities and there exist vital information/knowledge gaps. The present effort to model the global forests under three broad categories (tropical, temperate and boreal forests) and not including certain factors (natural hazards, drought, fire disturbance, biomass removals etc.) simplifies such complexity. The effort to make use of field observations and comparison with equivalent natural forest to calibrate the model with regard to biomass productivity has enhanced the value of the study results. The extended model is able to simulate observation-consistent results like plantations showing higher growth rates at least during the first 50 years. Despite major limitations such as only 3 species PFT, one each for tropical, temperate and boreal forests selected to simulate carbon uptake, harvest set to zero and omitting CO2 fertilization, the study still presents an opportunity to advance our capability to assess forestry-based options to mitigate climate change. The limitations quoted by the study bring focus on the areas for future advancement; most important among which is the characterization of PFTs for more number of common plantation species from the three zones. This study reports substantial work in the field of modeling for assessing attributes of forest plantation such as biomass (Carbon) accumulation. I find the methodology appropriate and results useful for publication. Study is likely to encourage more research work pertaining to forests as land use in general and plantations in particular.

*We thank the reviewer for the praise.*

Authors may however consider revising the zone and carbon density color-coding in figure-1 for easy comprehension.

*Indeed the color-coding for this figure made it a bit hard to read. We modified it as follows.*

[Figure]

Supplementary information may include methodology steps pertaining to selection of the representative species for the three zones and details of the location and source of observation data

*This information was indeed missing from the manuscript. For the calibration we required observations for fast-growing forest plantation species, in the form of time series for a sufficiently long period to assess the growth behaviour on time scales relevant to this study—at least 50 years. As discussed in section 2.3.1 of the manuscript, we found very few datasets in the literature which met these constraints. Therefore, the selection of data was not done according to a predefined set of criteria, but in a more ad hoc fashion. For the tropical FPFTs, the compilation of biomass growth presented by Brown et al. (1986) was very useful. For the temperate and boreal FPFTs we found no such compilations, hence we selected datasets for typical plantation species for wood production, poplar (populus spp.) and Scots pine (Pinus Sylvestris), respectively.*

*The first paragraph of section 2.3.2 has been modified to explain these points:*

*"Time series of stand-level stemwood C were collected from various sources in the literature. We required observations in the form of time series for a sufficiently long period to assess the growth behavior on time scales relevant to this study—at least 50 years. Due to limited data availability (see section 2.3.1), a rigorous data-selection procedure was not possible; hence the observations were collected in an ad hoc fashion. For the tropical FPFT we used data from Brown et al. (1986), who derived time series of stemwood biomass for several species and species groups for tropical forest plantations. For the temperate and boreal FPFTs, no such compilations were available, hence we used datasets for typical plantation species for wood production. Observations for natural poplar (populus x euramericana) forests were taken from Cannell (1982) for the calibration of the temperate FPFT. For the boreal FPFT, we used data for Scots pine (Pinus Sylvestris) plantations from Vanninen et al. (1996). Outliers in the observations were removed using Hampel filtering (Pearson, 2002). The data are depicted in Figure 2."*

Recommended for publication.